# The Role of Graciloplasty in the Treatment of Obstetric Anal Sphincter Injury with Subsequent Fecal Incontinence and Recurrent Low Recto-Vaginal Fistula: A Case Report

**DOI:** 10.3390/reports8010011

**Published:** 2025-01-20

**Authors:** Alessandro Bergna, Jacques Megevand, Giacomo Mori, Leonardo Lenisa, Andrea Rusconi

**Affiliations:** 1Department of General Surgery, Humanitas San Pio X, 20159 Milan, Italy; 2Department of Clinical-Surgical, Diagnostic and Pediatric Sciences, University of Pavia, 27100 Pavia, Italy; 3Department of General Surgery, Humanitas Clinical and Research Center, 20089 Rozzano, Italy

**Keywords:** ano-vaginal fistula, recto-vaginal fistula, obstetric anal sphincter injury, fecal incontinence, graciloplasty, case report

## Abstract

**Background and Clinical Significance**: Recto-vaginal fistulae (RVF) and fecal incontinence (FI) pose significant challenges for colorectal surgeons. Various therapeutic options have been proposed for each condition over time. Despite its procedural complexity and the risk of complications, graciloplasty remains a viable therapeutic option for both conditions, with favorable long-term results. To our knowledge, this is the first report of a case where the need to treat both conditions concurrently arose. **Case Presentation**: We report the case of a 54-year-old woman with severe FI and repeatedly operated on recurrent recto-vaginal fistula. The patient underwent graciloplasty to provide healthy tissue with an adequate vascular supply to both enhance the healing process of the fistula and reshape the anal canal with a circular muscular structure. Following the procedure, the patient experienced prompt symptom resolution and good clinical and functional recovery at a 1-year follow-up evaluation. **Conclusions**: This case report highlights the safety and effectiveness of an overlooked procedure for the treatment of large sphincter defects and concurrent recto-vaginal or recto-vaginal tears.

## 1. Introduction and Clinical Significance

Low recto-vaginal fistula (RVF) is a rare condition that presents a significant challenge for surgeons [1]. Fecal incontinence (FI) may coexist following obstetric injuries. Although nowadays there are various surgical techniques for the management of both RVF and FI, their long-term outcomes are often inconsistent, particularly in patients who have undergone previous multiple repair attempts. Reconstruction of the integrity of the recto-vaginal space and anal sphincter repair are considered primary targets for both RVF and FI treatments [2,3]. However, when both conditions co-exist, restoring normal anatomy becomes critical for enhancing functional outcomes. In such cases, graciloplasty is one of the most recommended procedures [2,4]. This technique offers the unique advantage of simultaneously interposing new vital tissue between the rectal and vaginal layers, thus supporting the closure of the recto-vaginal defect with healthy tissue and creating a neo-sphincter-like structure that improves fecal continence over time. However, due to the rarity of RVF and FI, and their even rarer co-occurrence, no standard of care has been established for these conditions. We report the case of a female patient who developed both RVF and FI after multiple pregnancies and had experienced failed repair of obstetric anal sphincter injuries (OASIS).

Clinical Significance: To our knowledge, this is the first report of a case where the need to treat both conditions concurrently arose. Our experience highlights the safety and the efficacy of graciloplasty for the treatment of extensive sphincter and perineal injuries, thus providing functional recovery and improving the patient’s quality of life.

## 2. Case Presentation

The this case report was written following the CARE Guidelines [5]. A 54-year-old woman with a history of three cesarean sections and three vaginal deliveries was referred to our institution (Humanitas San Pio X, Milan, Italy) for an outpatient evaluation in February 2023. The last labor (2013) was precipitous, resulting in grade IIIC OASIS. The patient subsequently underwent immediate perineorrhaphy. Following this, she developed an RVF and underwent three surgical repair attempts during the following year, with overlapping sphincteroplasty and multi-layered closure of the fistula. Although her symptoms initially improved, the patient afterward developed FI, urinary urgency, recurrent urinary tract infections, and reported the passage of gas through the vagina. Clinical examination revealed a severe anoperineal lesion, and anoscopy identified an anterior ano-vaginal fistulous tract. At the time of the first evaluation, the patient’s Cleveland Clinic Incontinence Score (CCIS) was 15 [6]. We used the Fecal Incontinence Quality of Life (FIQoL) questionnaire to assess the impact of the patient’s condition on her everyday activities (Table 1) [7].

Endoanal ultrasound (EAUS) demonstrated an anterior lesion of the internal anal sphincter extending 154° at the distal middle level, along with a lesion of the external anal sphincter covering 145° and a full-thickness interruption of the anterior anal canal wall (Figure 1).

These findings confirmed the presence of an 8.5 mm low RVF. Additional assessments, including anal manometry, revealed impaired voluntary contraction and rectal hypersensitivity. In April 2023, the patient underwent graciloplasty. Surgery was performed in the lithotomy position, under spinal anesthesia. A urinary catheter was installed at the beginning of the procedure. After the exploration of both the anal and vaginal canals, the fistula tract was found and excised (Figure 2).

Subsequently, the gracilis muscle was harvested using the index finger to complete its detachment from the right thigh. The neurovascular pedicle, which was found approximately 8 cm from the proximal insertion at the pubic bone, was preserved. A small incision was then performed at the level of the internal tibial tuberosity, where the distal tendon of the gracilis was cut (Figure 3).

Two bilateral incisions were performed at 1.5 cm from the anal verge. The muscle was then transposed to the perineal region, wrapped around the anus, and its tendon was anchored to the periosteum of the right ischial tuberosity (Figure 4). The RVF was simultaneously closed using the gracilis muscle flap, ensuring healthy tissue coverage.

The total operative time was 85 min. The postoperative days were uneventful, with progressive recovery. The urinary catheter was removed on postoperative day (POD) 2, and the patient was discharged on POD 3. Fifteen days after surgery, the patient reported near-complete resolution of her symptoms, with a CCIS of 5, and significant improvement in her quality of life. This improvement remained stable at the 3-, 6-, and 12-month follow-up evaluations (CCIS 5). At the 12-month outpatient evaluation, the total FIQoL was 98 (Table 1). The patient expressed high satisfaction with the therapeutic approach.

## 3. Discussion

RVF and FI are two of the primary indications for performing graciloplasty. However, its use remains controversial in the scientific literature due to its invasive nature and the uncertainty regarding long-term outcomes. Therefore, its indication remains limited to select cases with extensive sphincter damage. RVF represents one of the most challenging situations in colorectal surgery, given the diversity of available surgical techniques and their variable success rates. Pinto et al. (2010) and Corté et al. (2015) emphasized that RVF repair remains challenging, regardless of the technique, but concluded that no single approach can be considered as the benchmark procedure [8,9]. Instead, both studies advocated for patient-specific treatment plans based on the characteristics of the fistulae. According to the American Society of Colon and Rectal Surgeons (ASCRS) guidelines, the treatment of RVF should focus on both functional and anatomical restoration, thus relying on advanced techniques, like graciloplasty, when first-line repairs do not achieve good outcomes [8]. Maeda et al. (2023) assessed the effectiveness of different approaches for RVF repair, emphasizing the critical role of vascularized tissue in improving surgical outcomes and the success rate of surgical procedures for the treatment of refractory fistulae [10]. These findings support the rationale for using graciloplasty, as this technique involves the transfer of muscle with an intact neurovascular supply. A systematic review by Garoufalia et al. (2023) further highlighted the efficacy of graciloplasty for the treatment of complex perineal fistulae, with reported success rates exceeding 70% after follow-up periods of two years or more [11]. Additionally, patients treated with graciloplasty showed significant reductions in their CCIS, reflecting improved continence and good quality of life post-procedure. These results are consistent with the growing evidence supporting graciloplasty as an effective treatment for RVF, particularly in cases where previous repair efforts have been unsuccessful [11,12]. The creation of a neo-sphincter-like structure using the gracilis muscle flap can also restore continence by reshaping the anal canal and compensating for the loss of functional sphincter muscles. The success of graciloplasty in treating FI, especially in the presence of significant sphincteric damage, is supported by studies demonstrating favorable functional outcomes and sustained improvement in quality of life. However, it is essential to recognize that graciloplasty is a technically demanding procedure and carries the risk of long-term complications, such as muscle atrophy and functional failure, particularly in cases with extensive tissue damage [12,13]. Therefore, careful patient selection is critical to maximizing the benefits of the procedure and minimizing the potential for adverse outcomes. Our patient underwent multiple previous unsuccessful repairs of the fistula and the persistence of sphincteric damage led to the development of fecal incontinence. In this scenario, graciloplasty was the only current technique that combined the advantage of neo-sphincter reconstruction with the need to interpose vital tissue in the recto-vaginal septum. The dual benefit of this technique led to functional restoration and improvement of the patient’s quality of life. According to early CCIS measurement, fecal continence was restored quickly and maintained over time, with no early or late complications reported. As a side note, we considered gracilis muscle dynamization for the treatment of this patient to enhance and maintain the functional outcomes over time. However, currently, in our country, the electrodes required to perform the gracilis muscle’s stimulation are not marketed. For this reason, and given the good outcomes observed at follow-up, it was decided to postpone gracilis muscle stimulation to a later stage.

## 4. Conclusions

Graciloplasty is a reliable but technically demanding surgical procedure that can lead to surgical complications. For this reason, nowadays, graciloplasty should only be performed by experienced surgeons in the treatment of selected patients in referral centers. In our experience, graciloplasty was found to be safe and effective in the treatment of FI in the particular case of a patient affected by both sphincter damage and RVF.

## Figures and Tables

**Figure 1 reports-08-00011-f001:**
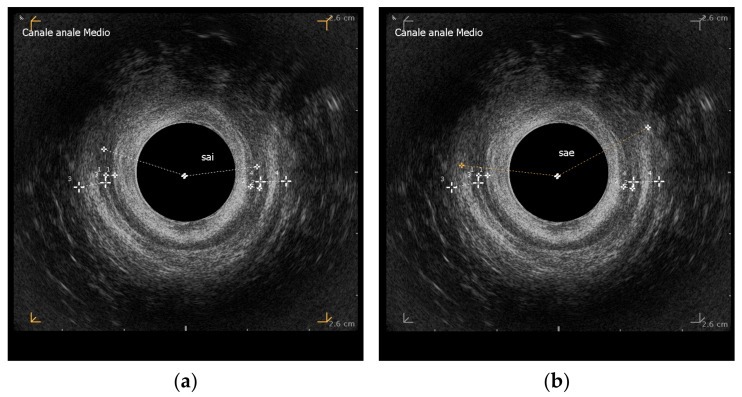
EAUS for the assessment of the anal sphincter complex. (**a**) Internal anal sphincter anterior defect of 154°; (**b**) external anal sphincter anterior defect of 145°.

**Figure 2 reports-08-00011-f002:**
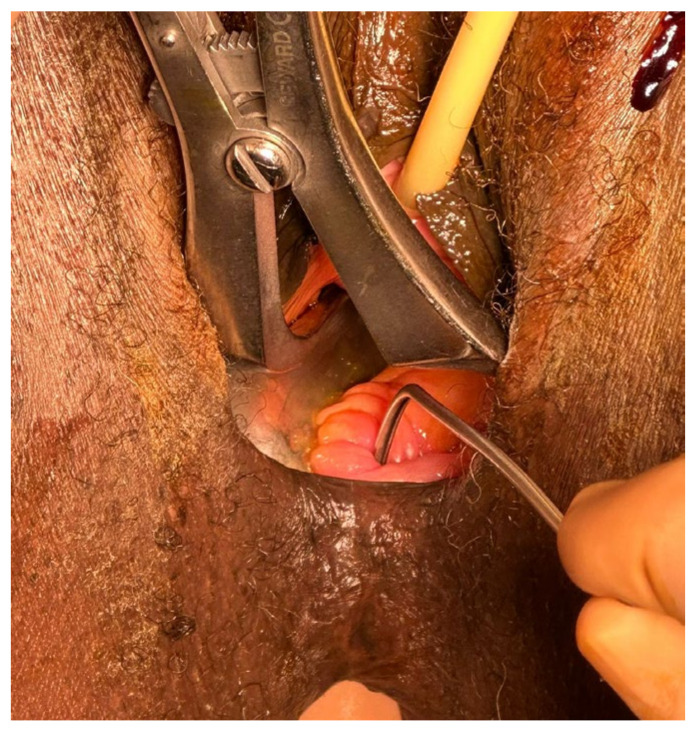
Ano-vaginal fistulous tract identification.

**Figure 3 reports-08-00011-f003:**
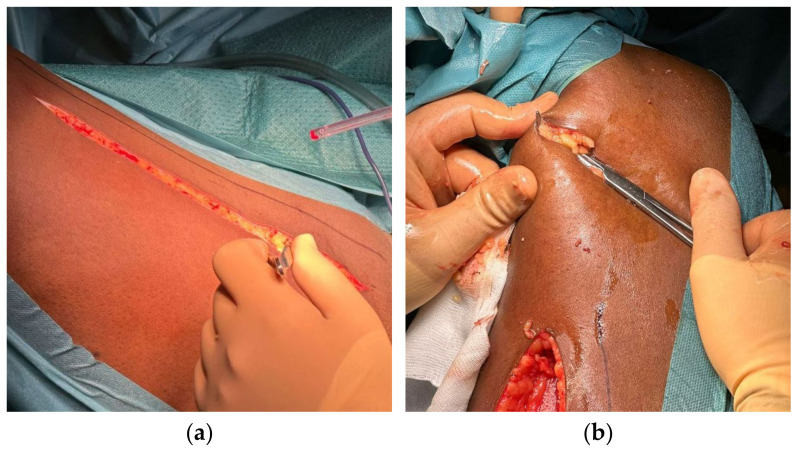
Dissection of the right gracilis muscle. (**a**) Skin incision; (**b**) harvest of the distal tendon of the muscle.

**Figure 4 reports-08-00011-f004:**
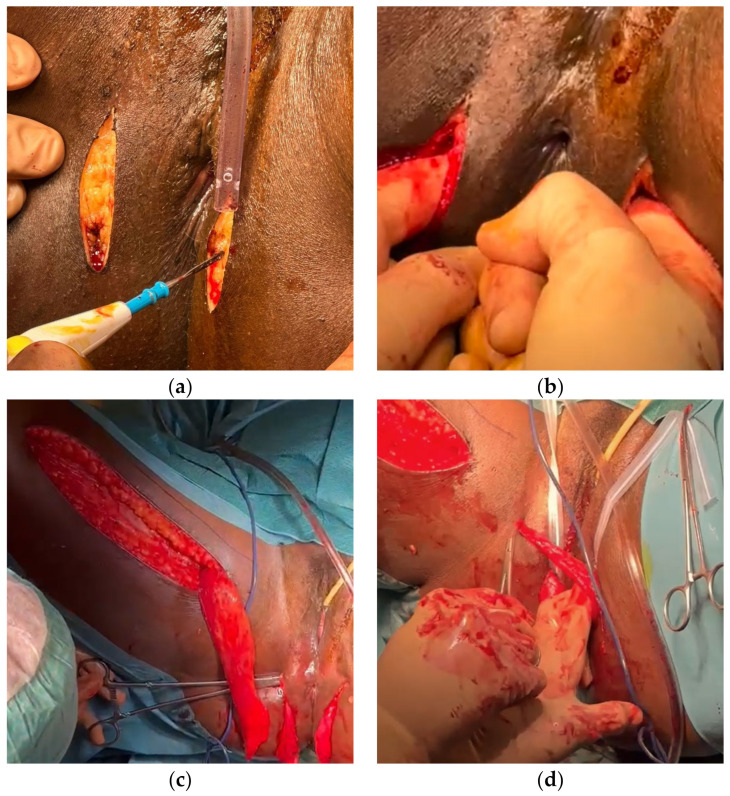
(**a**) Skin incisions in the perianal region; (**b**) Creation of a circumferential tunnel around the anus via blunt digital dissection; (**c**) transposition of the right gracilis muscle; (**d**) wrapping of the muscle around the anus.

**Table 1 reports-08-00011-t001:** FIQoL score at the first evaluation and at a 1-year follow up outpatient evaluation.

FIQoL Score	First Evaluation	1-Year Follow-Up
Total	46	98
Lifestyle	17	32
Copying\Behavior	14	30
Depression\Self Perception	12	26
Embarrassment	3	10

## Data Availability

All data underlying the findings are fully available by contacting the corresponding author.

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
