# Peer review of "The Role of Graciloplasty in the Treatment of Obstetric Anal Sphincter Injury with Subsequent Fecal Incontinence and Recurrent Low Recto-Vaginal Fistula: A Case Report"

_reports, 2025, doi:10.3390/reports8010011_

Round 1

Reviewer 1 Report

Comments and Suggestions for Authors

It is an interesting case report demonstrating the use of graciloplsdty in managing both a low recto signal fistula as an interposition grace and an external sphincter defect causing incontinence. However, the authors should have also discussed the facts that for a low rectovaginal fistula a simple repair in layers will suffice. The external anal sphincter defect would consequently have been repaired by an overlapping sphincteroplasty. The morbidity is less and the long term outcome is better than a graciloplasty for sphincter repair. A gracilopasty is useful following an additional electrostimulation because of the high long term failure rate of a simple graciloplasty. The striated muscle of the external anal sphincter has a greater proportion of type 1 fibres which are slow twitch fatigue resistant muscle fibers that derive their energy chiefly from aerobic metabolism and suitable for anal sphincter in function compared with the gracilis muscles. Thus , the need fir the conversion of the predominant fast twitch, fatigue prone, anaerobic dependent  type 2 fibers in Gracilis to type  1 fibres. The morbidity of the electro stimulated graciloplasty due to infection is also significant. Thus there is no real novelty in the case report. Long- term follow up is required to confirm the efficacy of this approach and procedure. 

Author Response

Dear Reviewer,
Thank you for your time and consideration. We greatly appreciate your comments and contribution for the improvement of our manuscript.

Comment 1: However, the authors should have also discussed the facts that for a low rectovaginal fistula a simple repair in layers will suffice. The external anal sphincter defect would consequently have been repaired by an overlapping sphincteroplasty. The morbidity is less and the long term outcome is better than a graciloplasty for sphincter repair. 

Response: As stated in lines 54-55 ("Following this, she developed a RVF and underwent three surgical repair attempts during the following year") the patient underwent previous simple repairs in layers: this is the reason why the repair of such a fistula required the interposition of a thicker tissue with the need to perform an "advanced repair". We clarify the fact that this kind of repair was performed by rephrasing those lines " Following this, she developed a RVF and underwent three surgical repair attempts during the following year with overlapping sphincteroplasty and multi-layered closure of the fistula"

Comment 2: The external anal sphincter defect would consequently have been repaired by an overlapping sphincteroplasty.

Response 2: As stated above, sphincteroplasty was previously performed in this patient in another center, and it failed. A re-do sphincteroplasty should generally be avoided according to ASCRS 2023 guidelines. Moreover, the presence of a large sphincterial defect (>120°) in a pluri-operated patient would have made it difficult to expose the retracted external anal sphincter and use the muscle to cover such a big defect.  

Comment 3:  A gracilopasty is useful following an additional electrostimulation because of the high long term failure rate of a simple graciloplasty. The striated muscle of the external anal sphincter has a greater proportion of type 1 fibres which are slow twitch fatigue resistant muscle fibers that derive their energy chiefly from aerobic metabolism and suitable for anal sphincter in function compared with the gracilis muscles. Thus , the need fir the conversion of the predominant fast twitch, fatigue prone, anaerobic dependent  type 2 fibers in Gracilis to type 1 fibres. The morbidity of the electro stimulated graciloplasty due to infection is also significant.

Response: We agree that Sacral Neuromodulation (SNM) is an effective technique for the treatment of Fecal Incontinence also in presence of a large sphincter defect. However, in this case, SNM has the disadvantage to do not treat the fistula. In this scenario, the repair of the fistula should have been performed through sphincteroplasty that, as stated before (see comments 2), was not the best treatment choice for this case. This is the reason why we decided to do not perform SNM in this case.

Comment 4: Thus there is no real novelty in the case report. Long- term follow up is required to confirm the efficacy of this approach and procedure. 

Response: We agree that long-term follow up would be beneficial to further confirm the efficacy of this approach and procedure. However, we are not discussing if graciloplasty is (or is not) a techinique with good long-term outcomes: we are ackwnoledging that graciloplasty is feasible and safe in the treatment of both fecal incontinence and concurrent recto-vaginal fistula in a patient that previously underwent surgery for the same reasons without achieving good clinical and functional outcomes. Moreover, to our knowledge, this is the first case reporting the (successful) attempt to repair a rectovaginal fistula and treat a concurrent fecal incontinence in a previous operated patient with residual large sphincter defect.

Reviewer 2 Report

Comments and Suggestions for Authors

The paper is well presented and describes a very interesting, original case. What I appreciated the most is how you underline both the pros and the cons of the procedure, how you stress the need for enrolling selected patients and how you report the lack of a general consensus for such cases. All the information is presented in a clear language, even though minor revision of the English is needed.

Comments on the Quality of English Language

Even though the article is clearly presented and all the information is easy to understand, I have noticed some sentences which need to be revised, for example:

- Line 22: "... the treatment OF large sphincteric ...".

- Lines 81-83: the use of "BECAUSE" makes the sentence hard to understand.

- Line 113: I think you should write "DO NOT achieve" instead of "achieve".

- Line 145: "... the treatment OF selected patients ...".

For such examples, I would recommend a minor review of the language.

Author Response

Dear Reviewer,
Thank you for your time and consideration. We greatly appreciate your comments and contribution for the improvement of our manuscript.

We have revised grammar and typos as per your suggestions:

-Line 22: we added "OF"

-Lines 81-83: we removed "BECAUSE"

- Line 113: The sentence has been rephrased employing a negation

- Line 145: we added "OF".

A thorough revision of the English language was then performed.

Reviewer 3 Report

Comments and Suggestions for Authors

Please correct the description of case: the phrase "the patient was referred to our outpatient clinic" was repeated twice

Author Response

Dear Reviewer,
Thank you for your time and consideration. We greatly appreciate your comments and contribution for the improvement of our manuscript.

We have removed the repeated sentence, as per your suggestion. 

Round 2

Reviewer 1 Report

Comments and Suggestions for Authors

I never suggested Sacral nerve stimulation in this case but an electrically stimulated graciloplasty for the graciloplasty not to fail. The author should discuss this. Apart of infection and cost, studies of electro stimulated gracious muscle transposition have shown a 75% continence rate in early and longer term follow- up studies

Author Response

Comments 1: The author should discuss this. Apart of infection and cost, studies of electro stimulated gracious muscle transposition have shown a 75% continence rate in early and longer term follow- up studies.

Response 1: We indeed considered gracilis muscle dynamization for the treatment of this patient. However, currently, in our country, the electrodes required to enable gracilis muscle stimulation are not commercially available. For this reason, and given the good functional outcomes observed at follow-up, it was decided to postpone gracilis muscle stimulation to a later stage. A statement has been added on lines 140-145 in the revised manuscript.